# The Effect of Nitrogen Content on Archaeal Diversity in an Arctic Lake Region

**DOI:** 10.3390/microorganisms7110543

**Published:** 2019-11-08

**Authors:** Jinjiang Lv, Feng Liu, Wenbing Han, Yu Wang, Qian Zhu, Jiaye Zang, Shuang Wang, Botao Zhang, Nengfei Wang

**Affiliations:** 1College of Chemistry and Chemical Engineering, Qingdao University, Qingdao 266071, China; 2Key Lab of Marine Bioactive Substances, First Institute of Oceanography, Ministry of Natural Resources, Qingdao 266061, China; 3Department of Biochemistry and Molecular biology, School of Basic Medicine, Qingdao University, Qingdao 266071, China

**Keywords:** nitrogen content, geochemical factor, high-throughput sequencing, archaeal diversity and community composition, soils and lake sediments

## Abstract

The function of Arctic soil ecosystems is crucially important for the global climate, and nitrogen (N) is the major limiting nutrient in these environments. This study assessed the effects of changes in nitrogen content on archaeal community diversity and composition in the Arctic lake area (London Island, Svalbard). A total of 16S rRNA genes were sequenced to investigate archaeal community composition. First, the soil samples and sediment samples were significantly different for the geochemical properties and archaeal community composition. Thaumarchaeota was an abundant phylum in the nine soil samples. Moreover, Euryarchaeota, Woesearchaeota, and Bathyarchaeota were significantly abundant phyla in the three sediment samples. Second, it was found that the surface runoff caused by the thawing of frozen soil and snow changed the geochemical properties of soils. Then, changes in geochemical properties affected the archaeal community composition in the soils. Moreover, a distance-based redundancy analysis revealed that NH_4_^+^–N (*p* < 0.05) and water content were the most significant factors that correlated with the archaeal community composition. Our study suggests that nitrogen content plays an important role in soil archaeal communities. Moreover, archaea play an important role in the carbon and nitrogen cycle in the Arctic lake area.

## 1. Introduction

Because of the amplification effect of global warming in the Arctic, Arctic temperatures are rising faster than those in low latitudes [1], and the latest research shows that Arctic warming has intensified over the past decade [2]. Climate warming may increase the thawing of permafrost and lead to changes in the distribution and abundance of thaw lakes and ponds [3,4]. 

In addition to the changes in the landscape, the geochemical properties of the Arctic ecosystem could also change. These properties include the pH, transitions between aerobic and anaerobic conditions, and the content of carbon and nitrogen [5], which will significantly change the biota and the function of ecosystems and may reshape the feedback process in climate change [6]. However, how climate changes translate into alterations of ecosystem dynamics is not well understood [7,8]. Microbial communities driving biogeochemical processes underlying nutrient fluxes may play an important role in the feedback effect of climate warming [9,10]. Therefore, understanding the structure and function of Arctic microbial communities is of great significance in predicting the response of Arctic ecosystems to climate warming [11]. 

Moreover, archaeal populations are rich in cold and temperate environments [12,13], and archaea not only exist in extremely high-temperature and halophilic environments but also in soils, seawater, fresh water, lake sediments, and other environments [14]. Archaea play an important role in the nitrogen cycle [15]. Ammonia-oxidizing archaea (AOA) mainly belong to Thaumarchaeota, which are ubiquitous and abundant in oligotrophic marine waters, estuaries, and sediments [16,17]. AOA participate in the rate-limiting reactions of nitrification processes and have recently been found to be potentially important in nitrogen cycling in a variety of environments [15]. Methanogenic archaea belong to Euryarchaeota. Some methanogenic archaea contain nitrogen-fixing genes [18]. Moreover, *Methanobacterium ivanovi* has been demonstrated to be able to fix molecular nitrogen [19,20].

There is abundant evidence indicating that nitrogen is the major limiting nutrient in Arctic soils [21,22]. Over geological timescales, the N cycle is thought to have an effect on the global biogeochemical cycle of carbon and change the content of atmospheric CO_2_ [23]. 

The thawing of permafrost and subsequent hydrological changes have also been shown to increase the availability of dissolved organic and inorganic nitrogen [24,25]. Surface runoff is formed by the thawing of frozen soil, and glaciers can erode ammonium nitrogen and nitrate nitrogen in soil, thereby enriching nitrogen during water erosion [26,27]. These changes raised the net primary productivity via more rapid N mineralization and uptake [28], which could affect carbon dynamics by ultimately changing the metabolic efficiency of the microbiome [5].

In this study, we selected the London Island of Konsfjorden as a sample point, as this island has typical slope features. Previous studies mainly focused on the correlation between bacterial communities and environmental factors [29]. The aim of this research is mainly to study the influence of nitrogen content changes caused by running water erosion on archaeal communities.

## 2. Materials and Methods

### 2.1. Sampling Site Description and Sample Collection

The study area is located on the London Island of Konsfjorden, which is on the west coast of Spitsbergen in the Svalbard archipelago (Figure 1a). Surface runoff is formed by the thawing of frozen soil and accumulated snow flows down the hillside into the lake during the summer. Samples were collected at four sites in July 2016 during China’s Arctic expedition. Three soil sites and one sediment site (Hill, Up, Down, and Sedi; Figure 1b) were sampled from the surface (5 cm) near each other (about 1 m apart) in triplicate. Fifty grams of samples were collected using a sterile shovel and put directly into TWIRL’EM sterile sampling bags (Labplas Inc., Sainte-Julie, QC, Canada). A total of nine soil samples and three lake sediment samples were obtained. After collection, samples were placed in centrifuge tubes at −20 °C in the Yellow River Station (Ny-Aalesund, Norway) for about 20 days and then taken to the home laboratory in China by plane. During the flights, the samples were conserved in an incubator with frozen ice bags. In the home laboratory, soil samples were frozen at −80 °C until nucleic acid extraction.

### 2.2. Geochemical Properties of Soils and Lake Sediments

A total of seven soil geochemical properties were measured, including pH, water content, organic carbon (OrC), and four soluble nutrients (NH_4_^+^–N, SiO_4_^2−^–Si, NO_3_^−^–N and NO_2_^−^–N; Table 1). Soil pH was measured by adding 10 mL of distilled water to four grams of soil for pH measurement using a pH meter (PHS-3C, Shanghai REX Instrument Factory, Shanghai, China). Water content (10 g of each sample) was determined as the gravimetric weight loss after drying the soil/sediment at 105 ℃ until reaching a constant weight. OrC was measured according to the following procedure. After freeze-drying, the soil sample was ground into powder and reacted with 5 mL 10% hydrochloric acid for 12 h, subsequently rinsed with Milli-Q water for complete acid removal, and dried overnight at 50 °C to be analyzed in an element analyzer (EA3000, Euro Vector SpA, Milan, Italy). The soils used to determine the nutrients were also freeze-dried, ground using an Agate mortar, and then water was added at a ratio of 1:10 (g/mL). After shaking once every 4 h for 48 h, a nutrient auto-analyzer (QuAAtro, SEAL, Germany) was used to determine other physical and chemical properties at a relative standard deviation of <5%.

### 2.3. DNA Extraction, Polymerase Chain Reaction (PCR) Amplification, and Sequencing

#### 2.3.1. DNA Extraction and PCR Amplification

The total genome DNA from the samples was extracted using the CTAB/SDS method. The DNA was extracted from 0.25 g soil from each sample using a Power Soil DNA Isolation Kit (MOBIO Laboratories, San Diego, CA, USA) according to the manufacturer’s instructions. DNA concentration and purity were monitored on 1% agarose gels. According to the concentration, DNA was diluted to 1 ng/μL using sterile water. The V4 and V5 hypervariable regions of the archaea 16S ribosomal RNA gene were amplified by polymerase chain reaction (PCR) using the primers Arch519F (5′-CAGCCGCCGCGGTAA-3′) and Arch915R (5′-GTGCTCCCCCGCCAATTCCT-3′). All PCR reactions were carried out in 30 μL reactions, including 15 μL of Phusion^®^ High-Fidelity PCR Master Mix with a GC Buffer (New England Biolabs, Ipswich, MA, United States), 0.2 μM of forward and reverse primers, and 10 ng template DNA. The PCR amplification cycle was as follows: Initial denaturation at 94 °C for 4 min, followed by 30 cycles of denaturation at 94 °C for 15 s, annealing at 56 °C for 30 s, and elongation at 68 °C for 80 s, with a final extension of 72 °C for 10 min.

#### 2.3.2. PCR Product Quantification, Qualification, and Purification

PCR products were mixed with equal volume of 1X loading buffer (containing SYB green) and loaded onto 2% agarose gel for detection. Samples with a bright main strip between 400–450 bp were chosen for further experiments. PCR products were mixed in equidense ratios. Then, a mixture of PCR products was purified with a Qiagen Gel Extraction Kit (Qiagen, Hilden, Germany).

#### 2.3.3. Library Preparation and Sequencing

Sequencing libraries were generated using the TruSeq^®^ DNA PCR-Free Sample Preparation Kit (Illumina, San Diego, CA, USA) following the manufacturer’s recommendations, and index codes were added. The library quality was assessed on a Qubit 2.0 Fluorometer (Thermo Scientific, Wilmington, DE, USA) and Agilent Bioanalyzer 2100 system (Agilent Technologies, Palo Alto, CA, USA). Lastly, the library was sequenced on an Illumina HiSeq2500 platform (San Diego, CA, USA), and 250 bp paired-end reads were generated.

#### 2.3.4. Processing of Sequencing Data

Paired-end reads were assigned to samples based on their unique barcode and truncated by cutting off the barcode and primer sequence, using FLASH (V1.2.7) to merge the paired-end reads to get the Raw Tags. Quality filtering on the raw tags was performed under specific filtering conditions to obtain high-quality clean tags according to QIIME (V1.7.0); then, chimeras were detected by the UCHIME algorithm and against the Gold database. Finally, the chimeric sequences and lllumina adapters were removed. The resulting effective reads of all samples were clustered using the Uparse software (V7.0, http://drive5.com/uparse/), and the sequences were clustered into OTUs (operational taxonomic units) with a 97% identity. Meanwhile, the most frequent sequence for an OTU was selected as the representative sequence of the OTU. The species were annotated and analyzed with a representative sequence of OTUs using QIIME and the SSU rRNA database SILVA (V 128) to obtain taxonomic information and calculate the abundance at each classification level in all the samples. The raw reads were deposited into the NCBI Sequence Read Archive (SRA) database (accession number: SUB6251077).

### 2.4. Statistical Analyses

The OTU abundance information was normalized using a standard sequence number corresponding to the sample with the least sequences. A subsequent analysis of the alpha diversity and beta diversity were all performed based on the output normalized data. Alpha diversity was applied in analyzing the complexity of species diversity for a sample based on 6 indices: Chao1, Shannon’s index (H′), Pielou, Simpson, ACE, and Good’s coverage. All indices in our samples were performed on QIIME (V 1.7.0) and displayed with R software (V 3.6.0). A one-way analysis of variance (ANOVA) followed by Tukey’s HSD (honest significant difference) test was performed for the geochemical properties and diversity parameters of the samples to determine the level of significance using the Statistical Package for the Social Sciences (SPSS) software (V.17.0). The relationships among the archaeal communities in the 12 samples were analyzed by a hierarchical clustering analysis using the R (V.3.6.0) statistical software. An analysis of similarities (ANOSIM) test was performed to determine whether the four sampling sites had statistically significantly different archaeal communities by using QIIME (V.1.7.0) software. The abundance-based Bray–Curtis similarity coefficient was used to examine the dissimilarity of the different samples. The relevance of environmental factors in explaining the distribution patterns of archaeal communities in different samples was analyzed by a Bray–Curtis distance-based redundancy analysis (db-RDA) using the R (V.3.6.0) statistical software. The input data can be standardized by taking the logarithm and other methods to ensure that continuous variables are sufficiently normally distributed. A Monte Carlo permutation test was also performed to examine the relationship between the 7 geochemical properties and the archaeal community composition in this Arctic lake area. In order to study the phylogenetic relationship of the different OTUs and the different dominant species in the different samples, a cluster analysis (heat map) of the archaeal genera at different sampling sites was performed with the R (V.3.6.0) statistical software. The species accumulation box-plot and Venn diagram were also developed using the by R (V.3.6.0) statistical software A linear discriminant analysis effect size (LEfSe) method was used to identify the significantly different archaeal groups in the different sampling sites.

## 3. Results

### 3.1. Geochemical Properties of Soil and Sediment Samples

The experimental results showed that there were some differences in geochemical properties among the four sites. The sediment samples were quite different between the soil samples. Moreover, there was a particular regularity in the change of the geochemical properties along the path of the surface runoff. Firstly, the contents of organic carbon, NO_3_^−^–N, and NO_2_^−^–N gradually decreased, while the contents of NH_4_^+^–N and SiO_4_^2−^–Si showed an increasing trend. The highest values of NH_4_^+^–N and SiO_4_^2−^–Si were observed from the sediment samples (site Sedi), where the lowest values of NO_3_^−^–N, NO_2_^−^–N, and organic carbon were also detected. Interestingly, the highest values of NO_3_^−^–N and NO_2_^−^–N and the lowest values of SiO_4_^2−^–Si were detected at the Up site, where there are some lichens. Furthermore, the water content of the three soil samples was basically the same (from 8% to 16%). However, the water content of the sediment samples was high at 22%. The pH values of all samples ranged from 7.63 to 8.22, which means that the soils were weakly alkaline. On the other hand, the pH value of the Down site was relatively high. The SiO_4_^2−^–Si of the Down and Sedi sites was higher than that of the Hill and Up sites, while the organic carbon content was lower.

### 3.2. Archaeal Diversity and Community Composition

In the 12 samples, a total of 716,728 raw sequences and 669,370 clean reads were retained after a series of quality control measures, and 2976 OTUs (at a 3% evolutionary distance) were identified in this study. Then, we eliminated the bacteria OTUs through the reference database and obtained 344 archaea OTUs (Figure 3). Down.1 contains the fewest OTUs (42), while Sedi.2 contains the most OTUs (276). The Good’s coverage estimator of the OTUs in the samples ranged from 0.997 to 1 (Table 2), indicating that the sequences sufficiently covered most of the archaeal populations in all samples. The Chao and AEC values of the three soil sample sites varied considerably. The lowest value appeared at the Down site, which indicates that the Down site had low species richness. In contrast, the sediment samples had high species richness. The changing trend of Shannon and Simpson is the same; the value of the sediment sample was higher than that of the soil sample, and the three soil sample sites were basically the same, indicating that the sediment had high species diversity. The Pielou of the soil sample ranged from 0.309 to 0.450, which is lower than the Pielou of the sediment samples (0.529–0.705), indicating that the archaeal community evenness of the sediment samples was high. The Shannon (4.116), Simpson (0.851), Chao1 (280.41), and AEC values (284.337) of Sedi.2 were the highest value among the 12 samples. The OTU accumulation box-plot (Figure 2) is saturated with all 12 samples, indicating that the OTUs of the study sites are well represented by these samples.

A Venn diagram (Figure 3) demonstrates that the OTUs differed among the four sites. The number of site-specific OTUs ranged from 5 (Site Down) to 190 (Site Sedi). Only 38 in 344 OTUs were shared by all four sampling sites.

In order to study the effects of nitrogen content change on archaeal communities, the general archaeal community structure was first studied. At different taxonomic ranks, the soil sample sites can be significantly distinguished from the sediment sample sites. For example, at the phylum rank (Figure 4), all OTUs were mapped to 9 phyla; members of Euryarchaeota (44–63%), Woesearchaeota (3–10%), Bathyarchaeota (MCG, Miscellaneous Crenarchaeota Group) (2–8%), and Sm1K20 (1–6%) were significantly abundant in the sediment samples. Thaumarchaeota was distributed in all 12 samples and was the dominate phylum, with a content of 97–99% in the three soil sample sites (Hill, Up, and Down). At the genus rank (Figure 5), the community structures and dominant genera of the Archaea at different sampling sites were quite different. Diapherotrites were unidentified, Parvarchaeota were unidentified, and Woesearchaeota were unidentified; they are close to each other in the phylogenetic tree and are mainly distributed in sediment samples. Indeed, they all belong to the superphyla DPANN [30]. *Candidatus*_Methanoperedens, *Methanobacterium*, *Methanoregula*, *Methanosaeta*, Methanobacteriaceae unidentified, *Methanosarcina*, and *Methanomassiliicoccus* were mainly distributed in sediment samples; except for *Candidatus*_Methanoperedens, all of them were methanogenic archaea. The abundance of *Candidatus*_Methanoperedens and *Methanobacterium* were also relatively high in the sediment samples. Some groups with few studies, such as SM1K20 (unidentified), MCG (Bathyarchaeota) (unidentified), MCG (unidentified), *Candidatus* lainarchaeum were also mainly distributed in sediments. *Candidatus* Nitrososphaera were distributed in all sampling sites and were more abundant in the soil samples. Moreover, they were far away from other archaea in the phylogenetic tree.

Besides the community composition, we also focused on archaea with large differences in archaeal abundance. Based on the LEfSe results, 17 taxa showed a linear discriminant analysis (LDA) score greater than 4 (the cutoff for significance test) in the 12 samples (Figure 6). Among all the samples, we found that Thaumarchaeota had the highest LDA (5.55) score, which belonged to the Up site, followed by Euryarchaeota (LDA 5.43). In these 12 samples, there were three phyla with significant abundance differences, including Thaumarchaeota, Euryarchaeota, and Woesearchaeota (LDA score 4.48). The Sedi site had 13 taxa that were more abundant than those of other sites. Methanobacteria and Methanomicrobia were two classes with significant abundance differences among the sediments.

### 3.3. Correlations between Environmental Variables and Archaeal Community Structure

A distance-based redundancy analysis (db-rda; Figure 7) and Monte Carlo permutation test (Table 3) were performed to examine the relationship between the seven geochemical properties and archaeal community composition in the Arctic area. This analysis also revealed the top 4 genera in the 12 sites (Appendix A): Thaumarchaeota (unclassified), *Candidatus* Nitrososphaera, *Methanobacterium*, and *Candidatus* Methanoperedens. A combination of the seven environment factors showed a significant correlation with the archaeal community composition, which explained 69.92% of the archaeal community variation. Among the seven geochemical factors, water content (*r*^2^ = 0.6685, *p* < 0.05) and NH_4_^+^–N (*r*^2^ = 0.6897, *p* < 0.05) were the most important factors associated with archaeal community composition in the area. *Candidatus* Methanoperedens and *Methanobacterium* were positively correlated with NH_4_^+^–N, SiO_4_^2−^–Si, and Water content, while Thaumarchaeota (unclassified) and *Candidatus* Nitrososphaera were positively correlated with NO_3_^−^–N, NO_2_^−^–N, and organic carbon.

## 4. Discussion

Our research has illustrated how the composition and diversity of archaeal communities in a lake area of the London Island changes with geochemical factors. In the present study, we only examined 344 archaea OTUs (Figure 3). The diversity of archaeal communities is much less than that of bacterial communities [29]. However, we have found that archaea play an important role in the carbon and nitrogen cycle in the Arctic region, by analyzing archaeal communities. Thaumarchaeota was significantly abundant in the three soil-sample sites and also distributed in the sediment samples. Euryarchaeota, Woesearchaeota, and Bathyarchaeota were mainly distributed in the sediment samples (Figure 4).

On London Island, geochemical properties play an important role in the changes of archaeal community diversity in the soil. The combination of the seven geochemical factors showed a significant correlation with archaeal community composition. In this study, NH_4_^+^–N (*r*^2^ = 0.6897, *p* < 0.05) and water content (*r*^2^ = 0.6685, *p* < 0.05) showed the highest correlation with archaeal community composition in all samples (Table 3), and other geochemical factors had different effects on different archaeal communities except for the pH factor (Figure 7). The dominant genera of archaea on London Island were *Candidatus* Nitrososphaera, *Methanobacterium*, *Candidatus* Methanoperedens, and Thaumarchaeota (unclassified) (Figure 7). Thaumarchaeota (unclassified) mainly belongs to the SCG (Soil Crenarchaeotic Group), and *Candidatus* Nitrososphaera belongs to AOA, which can oxidize NH_4_^+^ into NO_2_^-^ and fix carbon dioxide [31,32]. Furthermore, NO_2_^-^ can be reduced to N_2_O, which is a greenhouse gas [33]. Oxygen is needed in the process of ammonia-nitrogen oxidation so that the number of AOA is absolutely dominant in surface soil samples and less dominant in sediment samples [34]. All of these observations are consistent with the experimental results. *Candidatus* Nitrososphaera is negatively correlated with water content and NH_4_^+^–N but positively correlated with NO_3_^−^–N and NO_2_^−^–N (Figure 7). In general, soil ammonia-oxidizing archaea play a more important role in low total ammonia concentration environments where they were below 15 μg NH_4_^+^–N (g dw. soil)^−1^ [13,35]. The total ammonia concentrations of London Island were relatively low, ranging from 0.5 to 4.1 NH_4_^+^–N (g dw. soil)^−1^ (Table 1). Therefore, the AOA on London Island may play an important role in the nitrogen cycle and may be affected by geochemical properties.

Methanobacteria and Methanomicrobia were two significant classes of abundance differences for methanogenic archaea in the sediments (Figure 6). The dominant genus, *Methanobacterium*, which belongs to Methanobacteria, is a kind of methanogenic archaea and is strictly anaerobic. The formation of an anaerobic environment in flooded soil could explain the positive correlation between methanogenic archaea and water content [36]. However, the current study cannot disclose whether the observed effects of the water regime on methanogenic archaea are predominantly caused by redox effects, substrate availability, or a combination of these two factors. Further studies are needed [36]. In this study, *Methanobacterium* also had a positive correlation with NH_4_^+^–N; however, Methanogenic archaea are sensitive to ammonia nitrogen, and high concentrations of ammonia nitrogen inhibit methane production [37,38]. We suggest that the low concentration of ammonia nitrogen is not enough for inhibition, and the symbiotic relationship of the other microorganisms leads to this result. For example, the latest research results show a potential syntrophic relationship between Woesearchaeota and methanogens [39]. More research is needed in the future.

*Candidatus* Methanoperedens is a kind of Anaerobic Methanotrophic Archaea (ANME). In this study, *Candidatus* Methanoperedens belongs to Methanomicrobia, which is classified as ANME-1. ANME can denitrify with CH_4_ as an electron donor and NO_3_^-^ as an electron acceptor. This process is named nitrate-/nitrite-dependent anaerobic methane oxidation (N-DAMO) [40,41]. Without this process, there would be an additional 10–60% of CH_4_ in the atmosphere [42]. In this study, *Candidatus* Methanoperedens is positively correlated with water content and NH_4_^+^–N but negatively correlated with Orc, NO_3_^−^–N, and NO_2_^−^–N (Figure 7). All of these observations are supported by previous studies. *Candidatus* Methanoperedens plays an important role in the carbon and nitrogen cycle on London Island.

On London Island, we focused on the influence of nitrogen content changes caused by running water erosion on archaeal communities. Because the sample sites were close to each other, and the soil properties were basically the same as those of the surrounding environment, we believe that the change in the trend of the geochemical factors was due to the erosion of melting water caused by the melting of permafrost and snow. The close proximity of these sample sites to each other could also lead to some limitations. Moreover, the generality of the study is partially limited by the single locale where the samples were taken. Although our use of only 12 sampling sites means that there are not enough data to determine what is happening widely in the Arctic during the process of climate warming, our suite of examined variables and statistical analyses provide an interesting case study of likely ecological changes that are occurring more broadly across portions of the Arctic. In this study, the contents of NO_3_^−^–N and NO_2_^−^–N gradually decreased, while the contents of NH_4_^+^−N and SiO_4_^2−^–Si showed an increasing trend. This trend was the same as that of some other research results [43]. A portion of NO_3_^−^–N comes from the enrichment of snow to the atmosphere [44]. Moreover, NO_3_^−^–N can accumulate more significantly in dry soil sites because of the nitrification of AOA. By contrast, NO_3_^−^–N is consumed as N-DAMO increases in the sediments. NO_2_^−^–N, as an intermediate product of the nitrification and denitrification process, has a limited existence time [45]. Diverse biotopes, metabolic complementation, and the effects of the seven geochemical properties lead us to conclude that NH_4_^+^–N significantly affects the soil archaeal communities in this study. NO_3_^−^–N and NO_2_^−^–N also indicate the distribution and composition of some archaeal communities. The dominant genera of archaea like *Candidatus* Nitrososphaera, *Methanobacterium*, and *Candidatus* Methanoperedens all play an important role in carbon and nitrogen cycles. Through our analysis of archaeal communities, we infer that the main function of water content is to form an anoxic environment; we will undertake further research by determining oxygen content. We will also measure methane or CO_2_ to study the actual effects of archaea on carbon and nitrogen cycles in the Arctic region. Small ponds are not usually considered in large-scale greenhouse gas (GHG) and global carbon-cycling studies because they cannot be seen with remote sensing tools [46,47]. There are also some other factors that should be analyzed in future studies. Some studies suggest that soil depth and salinity may influence the distribution and composition of archaeal communities [4,48]. The results of this study may also show that, through runoff, terrestrial archaea may be transported into the seawater and sediments along the flow path. Considering the effects of global warming, our study provides reliable data for lakes formed by melting snow and ice in parts of the Arctic region.

## Figures and Tables

**Figure 1 microorganisms-07-00543-f001:**
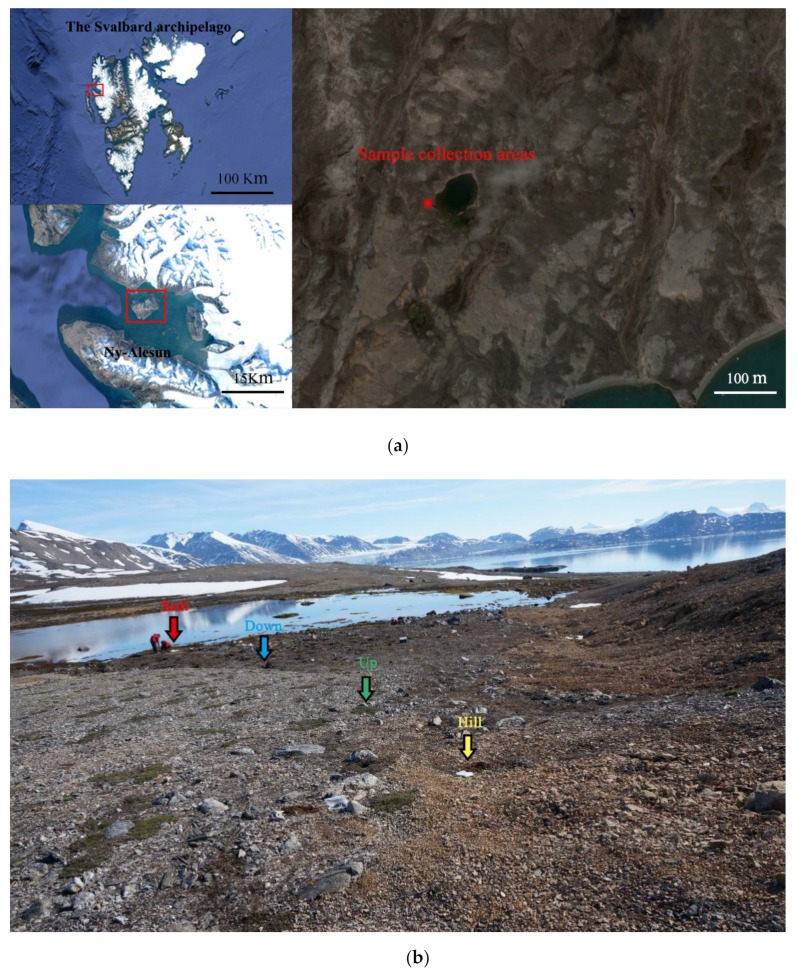
(**a**) Location; (**b**) the sampling sites from an Arctic lake area in the present study.

**Figure 2 microorganisms-07-00543-f002:**
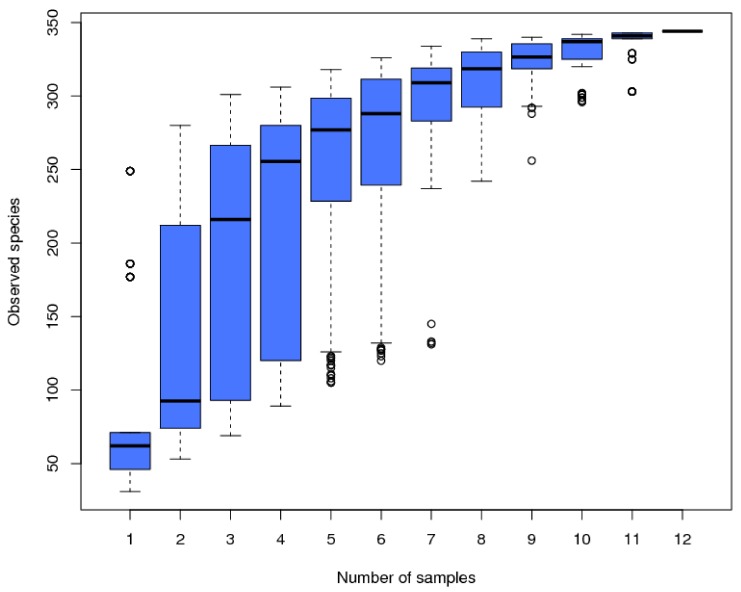
The species accumulation box-plot of the 12 samples.

**Figure 3 microorganisms-07-00543-f003:**
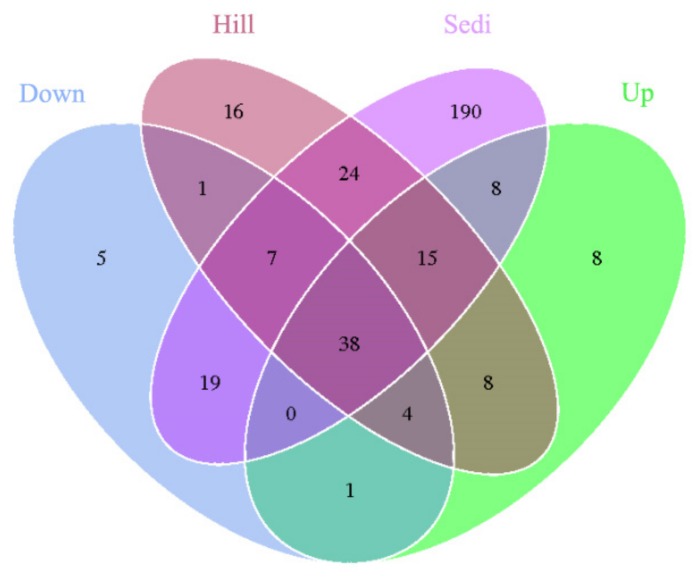
A Venn diagram displaying the degree of overlap of the archaea OTUs (at a 3% evolutionary distance) among the four sampling sites.

**Figure 4 microorganisms-07-00543-f004:**
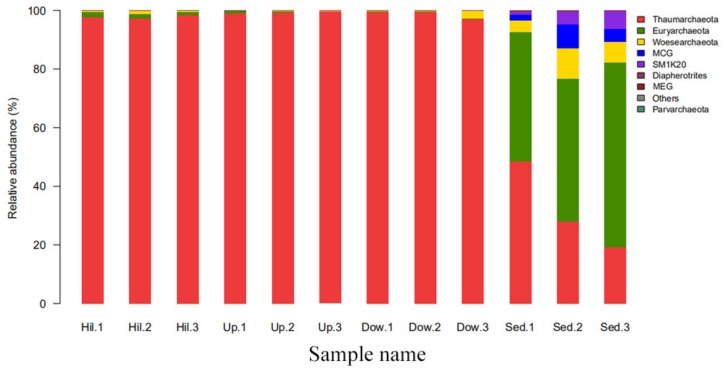
The top 10 abundance of the different phyla in the 12 soil samples of the present study.

**Figure 5 microorganisms-07-00543-f005:**
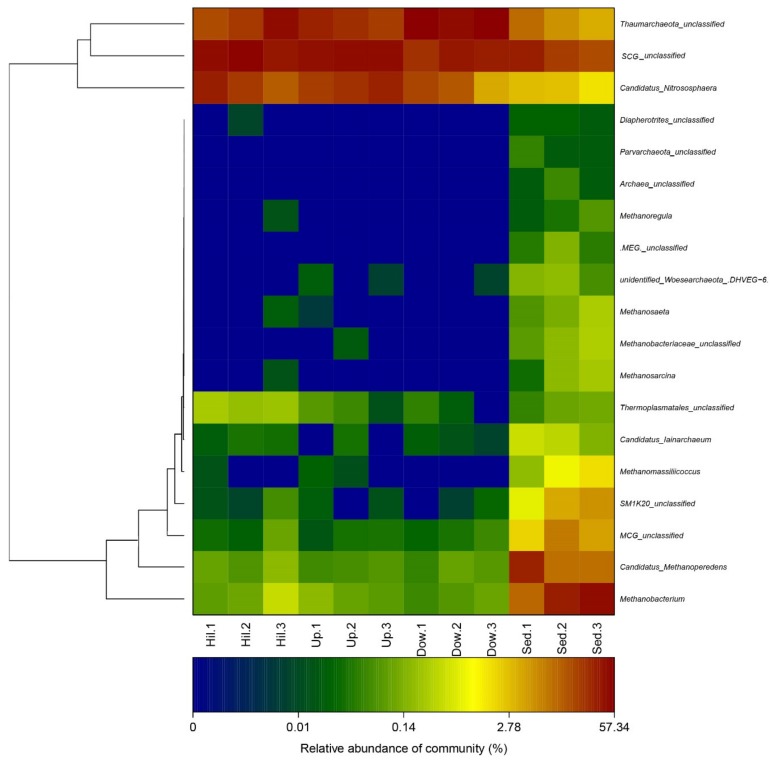
A heat map of the archaeal genus at different sampling sites in the Arctic lake area. A phylogenetic tree is on the left. The changes of the color gradient indicate the richness of the archaeal genus.

**Figure 6 microorganisms-07-00543-f006:**
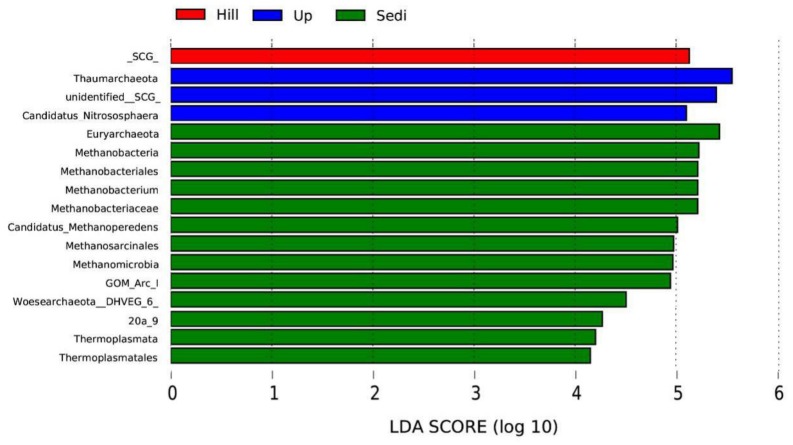
The taxa showing different relative abundances among the sites using the linear discriminant analysis (LDA).

**Figure 7 microorganisms-07-00543-f007:**
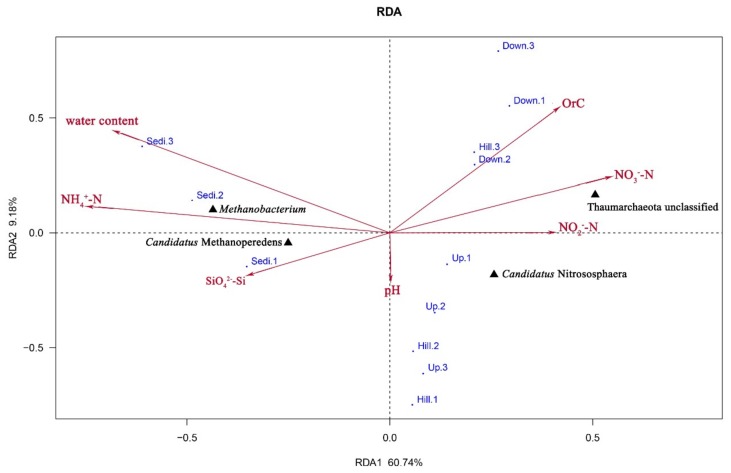
A distance-based redundancy analysis to show the correlations between the bacterial communities and environmental factors of the 12 samples from the four sampling sites.

**Table 1 microorganisms-07-00543-t001:** Geochemical properties of the 12 samples investigated in the present study.

Site	Sample	Water Content	pH	Organic Carbon (OrC) %	NH_4_^+^–N (μg/g)	SiO_4_^2−^–Si (μg/g)	NO_3_^−^–N (μg/g)	NO_2_^−^–N (μg/g)
Hill	Hill.1	0.14	7.63	1.18	0.94	2.25	0.06	0.04
	Hill.2	0.13	7.97	1.00	1.89	2.32	0.30	0.22
	Hill.3	0.14	8.03	1.06	0.50	1.85	0.62	0.04
	average	0.14 ± 0.01	7.88 ± 0.22	1.08 ± 0.09	1.11 ± 0.71	2.14 ± 0.25	0.32 ± 0.28	0.10 ± 0.10
Up	Up.1	0.16	7.62	1.53	2.85	1.43	0.62	0.23
	Up.2	0.11	7.68	1.16	1.85	1.75	0.52	0.15
	Up.3	0.10	7.85	0.12	0.75	2.19	0.03	0.04
	average	0.12 ± 0.03	7.72 ± 12	0.94 ± 0.73	1.82 ± 1.05	1.79 ± 0.38	0.39 ± 0.32	0.14 ± 0.10
Down	Down.1	0.11	8.13	0.25	1.15	4.36	0.20	0.03
	Down.2	0.11	8.22	0.21	0.93	3.11	0.20	0.03
	Down.3	0.08	8.11	0.16	1.39	4.50	0.44	0.06
	average	0.10 ± 0.02	8.15 ± 0.06	0.21 ± 0.05	1.16 ± 0.23	3.99 ± 0.77	0.28 ± 0.14	0.04 ± 0.02
Sediment	Sedi.1	0.18	7.91	0.11	3.27	3.11	0.05	0.01
	Sedi.2	0.23	7.84	0.51	4.08	4.29	0.01	0.01
	Sedi.3	0.19	7.90	0.31	3.44	3.06	0.01	0.01
	average	0.20 ± 0.03	7.88 ± 0.04	0.31 ± 0.2	3.60 ± 0.42	3.49 ± 0.70	0.02 ± 0.02	0.01 ± 0

**Table 2 microorganisms-07-00543-t002:** Summary data for the Miseq sequencing data from the 12 samples in the present study.

Sample Name	Raw Tag	Effective	OTUs	Shannon (H’)	Pielou (J’)	Simpson (1-D)	chao1	ACE	Good’s Coverage
Hill.1	29217	23991	75	2.491	0.426	0.773	83	90.254	0.999
Hill.2	21558	17420	74	2.629	0.450	0.796	65.882	71.368	0.999
Hill.3	25608	18117	87	2.248	0.385	0.707	122.667	123.325	0.998
Up.1	26854	23756	56	2.564	0.439	0.787	66	74.721	0.999
Up.2	26503	22461	60	2.392	0.410	0.775	54.667	56.963	0.999
Up.3	23635	20012	69	2.213	0.379	0.756	62.545	66.666	0.999
Down.1	28575	25156	42	1.803	0.309	0.632	36.25	36.296	1
Down.2	27944	22992	50	2.187	0.374	0.715	40.909	43.927	1
Down.3	26607	23313	65	2.114	0.362	0.643	53.077	57.351	0.999
Sedi.1	27538	20514	196	3.09	0.529	0.789	191.091	195.725	0.998
Sedi.2	31650	23542	276	4.116	0.705	0.851	280.41	284.337	0.997
Sedi.3	19396	14710	206	3.414	0.585	0.757	190.634	194.981	0.999

**Table 3 microorganisms-07-00543-t003:** A Monte Carlo permutation test of the relationship between environmental factors and archaeal community composition.

	RDA1	RDA2	*r* ^2^	Pr (>*r*)	
Water content	−0.963478	0.267788	0.6685	0.008	**
pH	0.123615	−0.992330	0.0209	0.908	
OrC	0.897715	0.440576	0.3345	0.149	
NH_4_^+^–N	−0.995861	0.090889	0.6897	0.008	**
SiO_4_^2−^–Si	−0.985292	−0.170879	0.1667	0.431	
NO_3_^−^–N	0.989857	0.142067	0.3876	0.108	
NO_2_^−^–N	0.999554	−0.029876	0.2080	0.362	

**correlation is significant at the 0.01 level.

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
