# Peer review of "The Effect of Nitrogen Content on Archaeal Diversity in an Arctic Lake Region"

_microorganisms, 2019, doi:10.3390/microorganisms7110543_

Round 1

Reviewer 1 Report

This is an interesting investigation on the diversity of archaea in soil and sediment samples from an Arctic Lake.

The following properties of the samples were determined: water content, pH, organic carbon, and concentrations of ammonia, silicate, nitrite and nitrate. The communities were compared on the basis of 16S rRNA gene sequences from libraries of the investigated samples. The sequences were classified in OTUs with a 3% sequence difference as limit and altogether almost 3000 OTUs were identified, whereof 344 represented archaea. The results show that the sediment samples differ from others by higher content of ammonia and lower ones of nitrate and nitrite and this correlates well with a significant increased diversity of the archaeal communities. However, other parameters could indeed also explain the difference between water covered sediment and soil samples and should be discussed in the manuscript.

The weakest point of the manuscript is the description of the diversity. The diversity is described superficially only on the basis of numbers of OTUs and their distribution among archaeota phyla. Completely missing is the identification of the individual OTUs, their relationship to known species (which could be shown in a phylogenetic tree and also a table) and their abundance in the samples. This information would show among others the abundance of archaea possibly involved in ammonia oxidation and relevant for the nitrogen cycling. The abundance of individual OTUs and their relationship to known representatives is important information to discuss the diversity of archaea on the nitrogen cycling and in dependence on the nitrogen content of the samples and should be included into the presentation of the data.

The formulations in many sentences give rise to misunderstanding and the English language needs a complete revision of the manuscript by a native English speaking person.

Author Response

Point 1

However, other parameters could indeed also explain the difference between water covered sediment and soil samples and should be discussed in the manuscript.

Response 1

We discussed other parameters to explain the differences between sediment and soil samples(174-187), such as SiO42--Si, Organic Carbon and Water content.

Point 2

The weakest point of the manuscript is the description of the diversity.

Response 2

A cluster analysis (heat map) of the archaeal genera at different sampling sites was performed to to describe species clustering and richness(Figure 5; 229-242), we analyzed the community distribution of Archaea at genus level. 

Point 3

The formulations in many sentences give rise to misunderstanding and the English language needs a complete revision of the manuscript by a native English speaking person.

Response 3

The manuscript has been checked for correct use of grammar and common technical terms.

Thank you very much for your suggestions. We fully realized the shortcomings in the manuscript and corrected them

Reviewer 2 Report

With increasing evidence that the Arctic climate change is dramatic, and that defrosting of the permafrost is leading to major changes in the edaphic and biotic components of these high latitude ecosystems, it is particularly useful to examine some of the effects on microbiota in soils and sediments, particularly archaeal communities that potentially can contribute significantly to atmospheric methane (a major greenhouse gas). This is an interesting study that contributes to this general theme. Though limited to one particular site in the Arctic, and four specific sampling sites along a terrestrial topographic relief, samples were taken in triplicate at all four sites. While the generality of the study is partially limited by the single locale where the samples were taken, the suite of variables examined, and the statistical analyses used, provide an interesting case study of likely ecological changes that are occurring more broadly across portions of the Arctic.  In this respect, I believe that the authors could expand the Discussion somewhat to consider the limitations of their one-site locale as well as the evidence that seems to support a wider applicability, including some of the relationships that they already include to prior published results. The authors have woven a nice narrative relating specific aspects of their work to prior findings, but it might be helpful to at least briefly acknowledge what might be some of the limitations of the one sampling site. Also, methodologically, some of the statistical analyses are parametric, and I wonder if the authors have adequately examined their continuous variables for normality as required. This is usually part of the statistical packages that they used. If these were done, a simple statement in the Methods that the continuous variables were verified to be sufficiently normally distributed to apply the parametric methods, would be sufficient to at least assure the reader.

With respect to the data in Table 2, it appears that all of the diversity indices are substantial, in addition to the specific comments the authors make about the highest values in Sedi.2 (lines 174-175). Perhaps in the Discussion (or Results) a brief comment acknowledging the relatively robust magnitude of the entire set of diversity values in Table 2 may be useful. Also, I wonder if the authors have considered reporting Pielou’s index (J’) of evenness in Table 2, that provides an estimate of the  ratio of the Shannon Index relative to the max possible Shannon value for each site (J’ = H’/H’max).  This would simply allow the reader to understand how closely the Shannon index for each site was to the max possible value. Also in Table 2, the headings should be Shannon, and Simpson (first letters capitalized).

There are a few changes in English that may improve the readability of some sentences as follows:

Lines                Recommended changes

41-42               “However, how climate changes could translate into alterations ----”

50                    “in the nitrogen cycle [15].”

62                    “---  thawing of frozen soil and glaciers can erode ammonium---”

98                    “--- were also freeze-dried, ground using ---------, and then water was added------”

The proper form of the verb is ground, not gounded, but it also would be useful to add what equipment was used to grind the soil sample as shown by the dashed lines.

115                  “---1X loading buffer (containing SYB green) and ----”

158                  “---and there was a particular regularity in the change of the geochemical properties along the path of surface runoff.”

171-172           “ --- the reference database and obtained 344 archaea OTUs (Figure 3).”

194-195           “ --- in sediment samples, Thaumarchaeota was distributed in all 12 samples and was the dominate phylum, with a content of 97-99% in the three soil----”

216-218           This sentence is not quite clear: “It was also shown the top 4 genera in the 12 sites, including Thaumarchaeota unclassified, Candidatus  Nitrososphaera, Methanobacterium , Candidatus  Methanoperedens.”  Where is this data? The beginning wording “It was also shown” is unclear, be specific where was this data shown?

254-255           “—furthermore, the NO2- can be reduced to N2O, which is --  -----”

260                  “ --- total ammonia concentration environments where they were below ----”

264                  “--- abundance difference classes of -----” [Au: is this what was intended to say?]

271                  “ ---- is also a positive correlation with NH4+-N; however, -----"

289                  “--- change in trend of the---

297                  “--- may lead us to conclude that NH4+-N is significantly affecting the----”

300                  “------carbon and nitrogen cycles.”

302                  “an anoxic environment; we will do----”

303                  “---- carbon and nitrogen cycles in the Arctic ---”

307                  “------- communities [4,48]. The results of this study may also show that through runoff, the terrestrial archaea may be transported into the seawater and sediments along its flow path.”

Author Response

Point 1

The generality of the study is partially limited by the single locale where the samples were taken, it might be helpful to at least briefly acknowledge what might be some of the limitations of the one sampling site.

Response 1

We expand the Discussion and consider the limitations of the one-site locale (344-350).

Point 2

Whether the author had examined their continuous variables for normality as required?

Response 2

We have made a supplementary explanation for this (146-148; 162-164). 

Point 3

With respect to the data in Table 2, it appears that all of the diversity indices are substantial, in addition to the specific comments the authors make about the highest values in Sedi.2 (lines 174-175). Perhaps in the Discussion (or Results) a brief comment acknowledging the relatively robust magnitude of the entire set of diversity values in Table 2 may be useful. Also, I wonder if the authors have considered reporting Pielou’s index (J’) of evenness in Table 2, that provides an estimate of the  ratio of the Shannon Index relative to the max possible Shannon value for each site (J’ = H’/H’max).  This would simply allow the reader to understand how closely the Shannon index for each site was to the max possible value. Also in Table 2, the headings should be Shannon, and Simpson (first letters capitalized).

Response 3

We have made a detailed supplement to other diversity indices in Table 2. In addition, we added the Pielou’s index of evenness and analyzed it(197-204). Moreover, some details have been corrected, the manuscript has been checked for correct use of grammar and common technical terms.

Point 4

216-218 This sentence is not quite clear: “It was also shown the top 4 genera in the 12 sites, including Thaumarchaeota unclassified, Candidatus  Nitrososphaera, Methanobacterium , Candidatus  Methanoperedens.”  Where is this data? The beginning wording “It was also shown” is unclear, be specific where was this data shown?

Response 4

We didn't upload the data at the first time, we want to submit it as supplementary Materials.

OTU ID Down.1 Down.2 Down.3 Up.1 Up.2 Up.3 Hill.1 Hill.2 Hill.3 Sedi.1 Sedi.2 Sedi.3
Archaea_unclassified 0.261984244 0.400866449 0.359542656 0.441581034 0.459106362 0.456378622 0.473866465 0.537689065 0.395795398 0.407259547 0.3945416 0.311879628
Thaumarchaeota_unclassified 0.541460809 0.453495213 0.57338024 0.33607088 0.280378825 0.214296822 0.173953035 0.223034962 0.462726922 0.10285234 0.059141494 0.038600914
Candidatus_Nitrososphaera 0.192415543 0.143017596 0.040633245 0.218181083 0.256872268 0.326628998 0.343090575 0.233139102 0.130221266 0.031149866 0.027927928 0.017252245
Methanobacterium 0.000534117 0.000748783 0.001114043 0.001820536 0.001019913 0.00088124 0.000919814 0.001177783 0.004690173 0.110889321 0.350768415 0.474003466
Candidatus_Methanoperedens 0.000489607 0.001016206 0.000820874 0.000566389 0.000631374 0.000777565 0.001028027 0.000743863 0.001876069 0.321951988 0.100158983 0.099968489

Thank you very much for your careful and rigorous suggestions. We have learned a lot.

Round 2

Reviewer 1 Report

The authors have carefully revised the manuscript, which is now considerably improved and may be published in the present form.